# Fine Mapping of *qSPJ_1* and Candidate Gene Identification for Soybean Seed Protein Content

**DOI:** 10.3390/plants14223525

**Published:** 2025-11-19

**Authors:** Jiayuan Chen, Jianguo Xie, Guang Li, Mingzhe Shen, Yuhong Zheng, Fanfan Meng, Xuhong Fan, Xingmiao Sun, Yunfeng Zhang, Mingliang Wang, Zhenyu Yang, Xin Xiong, Qiao Wang, Shuming Wang, Hongwei Jiang

**Affiliations:** 1College of Agriculture, Yanbian University, Yanji 133002, China; cjy1230311@163.com (J.C.); shen0224@ybu.edu.cn (M.S.); 2Jilin Academy of Agricultural Sciences (Northeast Agricultural Research Center of China), Changchun 130033, China; xiejg519@163.com (J.X.); lg-1207@163.com (G.L.); zhengyuhong520@163.com (Y.Z.); mengfanfan0720@163.com (F.M.); fxher@126.com (X.F.); xingmiaosun@163.com (X.S.); lzm1231111@163.com (Y.Z.); 15939325281@163.com (M.W.); zhenyu_yang2022@163.com (Z.Y.); 3College of Agriculture, Northeast Agricultural University, Harbin 150030, China; liuy34291@gmail.com; 4Institute of Industrial Crops of Heilongjiang Academy of Agricultural Sciences, Harbin 150086, China

**Keywords:** soybean seed protein, QTL fine mapping, candidate gene mining

## Abstract

Soybean (*Glycine max* L. Merr.) is rich in proteins, fats, and other nutrients, and the genetic improvement of soybean protein content has long been a key research focus in breeding programs. Based on the chromosome segment substitution line (CSSL) population, this study screened target lines within this population using genotypic and phenotypic information to establish an initial mapping population for soybean seed protein content. Through single-marker analysis, a quantitative trait locus (QTL) interval was mapped to the region between 26,705,080 bp and 33,180,908 bp on chromosome 16, designated as *qSPJ_1*. A secondary segregating population was constructed based on the initial mapping results for fine mapping, which narrowed the interval to 0.076 Mb. A total of 9 candidate genes were identified within this interval. By comparing amino acid and promoter sequences between the two parents, performing quantitative real-time PCR (qRT-PCR) analysis, and conducting haplotype analysis, *Glyma.16G165100* was preliminarily predicted as a candidate gene affecting soybean seed protein content. The single nucleotide polymorphism (SNP) variation sites in its promoter region were significantly associated with the variation in protein content in the resource population. This study provides important theoretical guidance for dissecting the genetic mechanism of soybean seed protein content and advancing its breeding improvement.

## 1. Introduction

### 1.1. Functions of Soybean Seed Protein

As a crucial natural macromolecule in nature [1], protein is an indispensable material basis for sustaining life activities. Soybean is an economic crop with high-quality protein [2,3], enriched in essential amino acids for humans [4]. The health benefits of soybean protein have been recognized by the U.S. Food and Drug Administration, and soybean products have been recommended by the World Health Organization as the best health food for the 21st century [5].

### 1.2. Influencing Factors of Soybean Seed Protein Content

The protein content in soybean seeds is influenced by multiple factors, which are mainly categorized into external and internal factors. External factors primarily include external environmental conditions such as geographical latitude, altitude, light, moisture, and temperature, as well as cultivation practices like irrigation and fertilization [6]. Relevant studies have shown that low CO_2_ concentration and short daylight hours are conducive to soybean protein formation. Excessively high or low temperatures are both unfavorable for protein accumulation, with the optimal temperature being 24 °C [7,8,9]. Additionally, moderate fertilization and irrigation levels contribute to increased soybean seed protein content [10]. Internal factors refer to the genetic background of soybean itself, such as additive effects, dominant effects, and epistatic effects among genes. Soybean seed protein content is a quantitative trait controlled by multiple genes, with a complex regulatory mechanism [11], and it generally exhibits a negative correlation with yield [12,13]. Traditional breeding methods are labor-intensive, time-consuming, and lack precision, making it difficult to break the unfavorable genetic correlation between soybean protein content and yield. Therefore, new technical approaches are needed to advance the genetic improvement of soybean seed protein content.

### 1.3. Research Progress on QTLs Related to Soybean Seed Protein Content

With the development of molecular biology, biotechnology has provided new strategies for crop breeding. Techniques such as marker-assisted selection, genomic selection, gene editing, and transgenesis can overcome the limitations of traditional phenotypic selection, enabling early, precise, and efficient genetic improvement. Researchers worldwide have conducted studies on the molecular genetic improvement of soybean seed protein content for several decades. QTL analysis has laid a crucial foundation for the development of molecular markers and the mining of functional genes. In 1992, Diers et al. [14] published the first study on QTL mapping for soybean protein content, which attracted widespread attention in the research community. Warrington et al. [15] constructed a recombinant inbred line (RIL) population using a high-yield, late-maturing cultivar and a high-protein, mid-maturing cultivar, identifying 4 QTLs associated with protein content. Whiting et al. [16] constructed a RIL population using the high-protein cultivar ACX790P and two high-yielding cultivars S18-R6 and S23-T5, and detected 14 QTL associated with protein content. Clevinger et al. [17] developed an RIL population by crossing with relatively high protein content PI399084 and with relatively low protein content PI507429. Based on 12,761 SNP markers, they identified QTL related to protein content on chromosomes 2 and 15 under four different environments, respectively. Jamison et al. [18] constructed a RIL population using high protein, low sucrose R08-3221 and high sucrose, low protein R07-2000. Genotypic data were generated via SoySNP6k chip analysis, leading to the identification of two protein content-related QTLs, which were localized on chromosomes 11 and 20, respectively. Kim et al. [19] crossed the soybean cultivar “Daepung” with the high-protein cultivar “GWS-1887” and detected a QTL associated with protein content at the genomic position Gm20_29512680. To date, the Soybase database (www.soybase.org) has recorded over 200 QTLs associated with soybean seed protein content [2].

### 1.4. Research Progress on Genes Related to Soybean Seed Protein Content

Notable progress has been made in mining genes related to soybean seed protein content in recent years. In 2020, Wang et al. [20] reported via population genetics that GmSWEET10a and GmSWEET10b underwent progressive variation and artificial selection: GmSWEET10a was strongly selected during soybean domestication, leading to larger seeds, increased oil content, and decreased protein content in cultivated soybeans; the domestication and selection of GmSWEET10b lagged behind that of GmSWEET10a. These two genes coordinately regulate seed size, oil content, and protein content, playing a key role in soybean domestication and improvement. In 2022, researchers [21] confirmed through transgenic experiments that GmST05—the ortholog of AtMFT in soybean, where AtMFT refers to Mother of Flowering Time from Arabidopsis thaliana—positively regulates soybean seed size; further studies revealed that GmST05 may affect seed oil and protein content by regulating the transcription of GmSWEET10a.In 2023, Qi et al. [22] identified and validated FA9 as a major gene controlling soybean seed oil content on chromosome 9 using phenomics, genomics, and gene editing; FA9 encodes a SEIPIN protein and also exerts pleiotropic effects on seed size and protein content. In 2025, Yang et al. [23] identified GmGASA12 on chromosome 8 as a key gene influencing soybean seed size and protein content; GmGASA12 encodes a gibberellin-responsive protein. Knockout of GmGASA12 synergistically increased water-soluble protein content and yield per plant, while expanding seed cells by 27.6%. The continuous identification and validation of functional genes provide a solid basis for the design-based breeding of soybean seed protein content and new insights into achieving simultaneous improvements in yield and quality.

### 1.5. Significance of This Study

Based on a genome-wide introgression line population containing genetic information from wild soybean, this study constructed an initial mapping population and a secondary segregating population for soybean seed protein content. Ultimately, a previously unreported QTL for soybean seed protein content, designated as *qSPJ_1*, was mapped on chromosome 16, with the gene conferring increased protein content derived from wild soybean. Additionally, potential candidate genes were predicted within this QTL region. This study provides new guiding information for the development of molecular markers related to soybean seed protein and offers a new reference for the application of wild soybean resources in soybean genetic improvement.

## 2. Results

### 2.1. Initial Mapping of Protein Quantitative Trait Loci (Protein QTLs)

Individual plants with homozygous backgrounds, minimal heterozygous segments, and few introgressed segments were selected from the CSSL population. Based on phenotypic data of protein content, the CSSL-646 population—with significant segregation and normal distribution of protein content—was selected as the initial QTL mapping population. The CSSL-646 population contained 7 heterozygous segments on chromosomes 8 (Gm08), 10 (Gm10), and 16 (Gm16), and 11 homozygous introgressed segments, with heterozygous and introgressed segments mainly concentrated on Gm16 (Figure 1).

The FOSS-1241TM analyzer was used to measure the seed protein content of 79 individual plants in the CSSL-646 population. The protein content ranged from 39.53% to 44.56%, showing significant segregation and normal distribution (Figure 2). Segregation Analysis (SEA) software [24] analysis indicated that the protein content trait in this population was controlled by a major gene with dominant effects, confirming it as a single-gene population (Table 1).

Based on whole-genome resequencing data, SSR primers were selected from public primer libraries: 6 polymorphic SSR primers were screened for the homozygous introgressed segments on Gm08, 28 for the heterozygous segments on Gm16, and 8 for the introgressed segments on Gm16. Forty extreme phenotypic individuals, including 20 high-protein individuals and 20 low-protein individuals, were selected from the CSSL-646 population for genotyping using Polyacrylamide Gel Electrophoresis. By comparing genotypic differences between extreme phenotypes, significant genotypic differences were observed on Gm16. Thus, the QTL associated with soybean seed protein content was initially mapped to the interval of 26,705,080 bp–33,180,908 bp on Gm16, designated as *qSPJ_1*.

### 2.2. Fine Mapping of Protein Locus qSPJ_1

Four individuals, namely 646−2−14, 646−2−15, 646−2−15, and 646−2−25, with heterozygous genotypes in the *qSPJ_1* interval were selected from the F_2_ generation and grown as single rows to construct the R1 population. The protein content of the R1 population showed a normal distribution (Figure 3). SEA software analysis was performed on the protein content phenotypic data of R1 (Table 2). Twenty high-protein and 20 low-protein individuals from R1 were selected for genotyping: the genotypes of high-protein and low-protein individuals on Gm08 were consistent with the SN14 background, excluding the influence of Gm08 on protein content. Marker densification was performed within the *qSPJ_1* interval, and single-marker analysis narrowed the interval to 1.0 Mb (32,403,844 bp–33,428,851 bp), flanked by markers BARCSOYSSR_16_1082 and BARCSOYSSR_16_1134.

Based on the genotyping results of the R1 population, four individuals—namely 646−2−15−14, 646−2−21−20, 646−2−21−23, and 646−2−21−24—with heterozygous target segments were selected, and their progeny (182 individuals) were used to construct the R2 population. The R2 population showed a distinct bimodal distribution of protein content. (Figure 3) Chi-square test using SPSS 26.0 software [25] indicated that the phenotypic segregation ratio conformed to the 1:3 Mendelian ratio for single-gene control. SEA software analysis confirmed that the R2 population was controlled by a major gene (Table 2). Genotyping of 40 extreme phenotypic individuals (20 high-protein, 20 low-protein) from R2 revealed segregation of the target segment into two sub-intervals: 0.076 Mb (32,403,844 bp–32,480,066 bp, flanked by BARCSOYSSR_16_1082 and BARCSOYSSR_16_1087) and 0.36 Mb (33,069,984 bp–33,428,851 bp, flanked by BARCSOYSSR_16_1123 and BARCSOYSSR_16_1134).

Individuals with heterozygous genotypes in the two sub-intervals and homozygous genotypes in other regions were selected, and 9 heterozygous individuals (W1–W9) were used to construct the R3 population, which is an residual heterozygous line (RHL) population. The R3 population showed significant segregation of protein content, with values ranging from 39.95% to 47.76%. Single-marker analysis of primers within the two sub-intervals revealed that four markers—namely BARCSOYSSR_16_1082, BARCSOYSSR_16_1083, BARCSOYSSR_16_1084, and BARCSOYSSR_16_1087—were extremely significantly associated with protein content. Thus, the qSPJ_1 interval was finally refined to 0.076 Mb (32,403,844 bp–32,480,066 bp) on Gm16 (Figure 4 and Figure 5).

### 2.3. Gene Annotation and Parental Sequence Alignment of Candidate Genes

Functional annotation of genes within the 0.076 Mb fine-mapped interval identified 9 candidate genes (Table 3).

The coding sequence (CDS) of candidate genes from SN14 and ZYD00006 were translated using the Expasy website (http://web.expasy.org/translate/), and their amino acid sequences were further compared. Only *Glyma.16G164900*, *Glyma.16G165000*, *Glyma.16G165100*, *Glyma.16G165300*, and *Glyma.16G165600* were found to have nonsynonymous mutations (Figure 6).

### 2.4. Quantitative Real-Time PCR Analysis (qRT-PCR Analysis)

To further validate the five candidate genes with non-synonymous mutations, two low-protein individuals—D646−1 and D646−2, which have the SN14 background in the target segment—and two high-protein individuals—D646−3 and D646−4, which have the ZYD00006 background in the target segment—were selected from the RHL population, with SN14 used as the control. qPCR analysis showed significant differences in the relative expression levels of *Glyma.16G164900* and *Glyma.16G165100* between high-protein and low-protein materials (Figure 7 and Figure 8).

### 2.5. Analysis of Promoter Elements of Candidate Genes

Using the Plant CARE website, the 3000 bp upstream sequences of *Glyma.16G164900* and *Glyma.16G165100* from the sequencing data of the two parents (SN14 and ZYD0006) were extracted and compared for analysis. The promoter regions of the candidate genes *Glyma.16G164900* and *Glyma.16G165100* both contain a large number of core promoter elements around transcriptional promoters (TATA-box) and cis-elements that function cooperatively in the promoter and enhancer regions (CAAT-box). Additionally, variations in varying degrees exist in the promoter regions of these two genes between the parents, which may play a regulatory role in gene expression (Figure 9).

### 2.6. Haplotype Analysis of Candidate Genes

Haplotype analysis of candidate genes *Glyma.16G164900* and *Glyma.16G165100* was performed in a resource population consisting of 350 soybean core germplasms (Appendix A). Whole-genome resequencing data were used for the analysis of these two genes. For *Glyma.16G164900,* two elite haplotypes were identified in the soybean resource population, with no significant difference in protein content between the elite haplotypes (Appendix A). For *Glyma.16G165100*, a total of four elite haplotypes were classified in the resource population (Figure 10 and Figure 11). Among them, the average protein content of Hap_3 and Hap_4 was significantly higher than that of Hap_1. Therefore, Glyma.16G165100 was inferred to be the candidate gene associated with the soybean protein content phenotype.Hap_3 included 350 accessions, with 19 from Heilongjiang, 8 from Jilin, 2 from Inner Mongolia, and 1 each from Xinjiang and Liaoning. Hap_4 contained 16 accessions, including 9 from Heilongjiang, 4 from Jilin, and 3 from Liaoning. These results indicated that soybean germplasms from Heilongjiang had greater advantages in high protein content among the elite haplotypes.

The main differential loci between Hap_3, Hap_4, and Hap_1 were concentrated in the promoter region. In order to further explore the association between the variation in the 3000 bp upstream regulatory region of the candidate gene Glyma.16G165100 and the protein content of soybean seeds, this study detected and analyzed the SNP variation information in the upstream regulatory region of the gene in 350 soybean genotype materials. The results showed that a total of 6 SNP loci were identified in this region; sequence alignment and element prediction showed that these SNP variations may lead to structural changes in the core cis-acting elements (TATA-box and, CAAT-box and ARE, etc.) in the regulatory region, which may regulate the expression level of Glyma.16G165100 gene by affecting the transcription initiation efficiency (Appendix A). Red indicates Hap_1, Green indicates Hap_2, Blue indicates Hap_3, Purple indicates Hap_4, In the labels, groups sharing the same letter indicate no significant difference in the average protein content, while groups with no common letter indicate a significant difference in the average protein content. 

## 3. Discussion

Wild soybean exhibits higher genetic diversity compared with cultivated soybean [1]. During long-term natural selection, wild soybean exhibits tolerance to abiotic stresses such as salinity-alkalinity, drought, aluminum toxicity, and phosphorus deficiency [26,27,28]. Wild soybean also harbors numerous quality-related genes, which are associated with traits such as protein content, vitamin, and amino acid [29,30]. In this study, a CSSL population derived from the wild soybean ZYD00006 and the cultivated soybean Suinong 14 was used as the basic material. This population has a Suinong 14 genetic background, and each line contains a small number of wild soybean gene segments, which induce phenotypic variation. Lines with fewer introgressed segments were screened from the CSSL population; through continuous screening and generation advancement, a mapping population for soybean seed protein content was constructed. This mapping population has a clean genetic background and low genetic “noise”, thus ensuring higher accuracy and precision of the mapping results. Furthermore, the introgression of genetic segments from wild resources enables the detection of elite allelic variations that are absent in cultivated varieties. In this study, the allelic genes contributing to increased seed protein content were derived from wild soybean resources, which provides an important reference for the utilization of elite alleles from wild soybean and is conducive to broadening the genetic basis of cultivated soybean varieties.

Previous studies have mapped multiple QTLs associated with soybean seed protein content on chromosome 16. Jun et al. [31] mapped two related QTLs using SSR markers, with interval lengths of 0.9 Mb (flanked by marker Satt287) and 1.7 Mb (flanked by marker BARC_025851), respectively. Among these QTLs, the one linked to marker BARC_025851 is close in position to qSPJ_1 detected in this study. Sonah et al. [32] identified one QTL that simultaneously controls protein and oil content in the interval of 4,183,401–4,523,670 bp using GWAS and GBS technologies. Li et al. [33] detected three QTLs associated with protein and oil content in the interval of ss715624939–ss715624938 using SNP markers. These QTLs do not overlap with the *qSPJ_1* locus mapped in this study; thus, this study identified a novel QTL for soybean seed protein content, with the elite allelic variation derived from wild resources. Furthermore, within the QTL interval, this study mined candidate genes and performed haplotype analysis of these candidate genes using soybean germplasm resources from different regions, ultimately obtaining elite haplotypes associated with soybean seed protein content. Therefore, this study will provide an important reference for marker-assisted selection breeding of soybean seed protein content.

In this study, through gene sequence alignment, gene expression level analysis, and haplotype analysis, the WD-40 repeat protein family gene *Glyma.16G165100* was preliminarily predicted to be a key candidate gene affecting soybean seed protein content. As one of the largest protein families in eukaryotes, the WD-40 repeat protein family is characterized by conserved repeat sequences of approximately 40 amino acids, with functions covering multiple key biological processes such as cell division, light signal transduction, cell cycle regulation, and growth and development [34]. Van et al. [35] reported that WD-40 proteins interact with TPR family genes through their repeat motifs. TPR family proteins exhibit specific interactions with ribosomal components [36]. Liu et al. [37] identified the WD-40 protein family gene SHREK1 in maize, which plays an important role in ribosome biogenesis and kernel development. Ribosomes are the “factories” for protein synthesis, enabling the continuous addition of amino acids through a cyclic reaction of “aminoacyl-tRNA entry → peptide bond formation → translocation”. When ribosomes move to the stop codons (UAA, UAG, UGA) of mRNA, protein synthesis enters the termination stage. Therefore, it is hypothesized that *Glyma.16G165100* indirectly affects soybean seed protein content by influencing ribosome synthesis, which requires further verification.

Soybean protein content is negatively correlated with oil content, yield, and ambient temperature [12]. Genetic improvement of protein content requires balancing the synergy and trade-offs among multiple traits, which is also a core challenge in soybean quality breeding [38,39,40]. This is because protein synthesis consumes large amounts of photosynthates and energy. For soybeans, the total amount of photosynthates is relatively fixed; if a large portion of photosynthates and energy is consumed for protein synthesis, the substances and energy available for other biological processes will decrease accordingly. Zhong et al. [39] found that the rhizobium-induced cle1a/2a (ric1a/2a) mutants exhibited a moderate increase in nodule number, balanced carbon allocation, and enhanced carbon-nitrogen acquisition capacity. The two ric1a/2a lines showed improvements in seed yield, protein content, and persistent oil content, indicating that gene editing toward optimal nodulation enhances soybean yield and quality. The candidate gene *Glyma.16G165100* predicted in this study has not been validated for its function through genetic transformation experiments, which represents a limitation. In subsequent studies, gene-edited mutants will be generated to confirm its gene function. In terms of gene function, *Glyma.16G165100* may indirectly affect protein synthesis by influencing ribosome synthesis; meanwhile, various enzymes involved in oil synthesis and yield formation are also synthesized by ribosomes. Therefore, editing this gene may enable the coordinated positive regulation of soybean quality and yield.

In this study, the planting location, cultivation method, and management mode of the QTL mapping population were standardized over 4 years, and QTL mapping was performed each year to reduce the interference of environmental heterogeneity on protein phenotypes. Secondary segregating populations were constructed in different years, and the same QTL was repeatedly detected in all cases, indicating that the QTL is stable. Furthermore, haplotype analysis of the candidate gene was conducted using phenotypic data from a 3-year germplasm population collected from other locations, and elite haplotypes were identified. This confirms the accuracy of the candidate gene prediction and its broad applicability to a certain extent. However, there are also some limitations, such as QTL mapping materials planted in a single location. The test year is limited. Moreover, the source of q PCR test materials is relatively simple. There is a lack of verification of the function of candidate genes in genetic transformation. These limitations also point out the direction for our future research.

## 4. Materials and Methods

### 4.1. Experimental Materials

#### 4.1.1. Construction of the CSSL Population

The genetic population used in this experiment was a CSSL population previously constructed from the Heilongjiang cultivated soybean variety Suinong 14 and the wild soybean accession ZYD00006 [41], where Suinong 14 served as the recurrent parent and wild soybean ZYD00006 as the donor parent. After the F_1_ progeny were obtained from the cross between the two parents in 2006, high-generation genetically stable lines were developed through continuous backcrossing and selfing. Through genotypic screening and phenotypic identification, a total of 220 stable lines were obtained, which fully cover the soybean genome.

#### 4.1.2. Construction of Mapping Populations

Based on whole-genome resequencing results, 23 individual plants with relatively homozygous backgrounds and few heterozygous segments were selected from the CSSLs population. In 2017, these individual plants were propagated into lines, and the CSSL-646 population was finally identified as the initial QTL mapping population according to the segregation pattern of protein content. According to the initial mapping results, in 2018, individual plants with heterozygosity in the initial mapping interval and homozygosity in other regions were selected from the CSSL-646 population to construct a secondary segregating population containing 99 plants, named R1. In 2019, based on the mapping results of R1, a RHL population with 182 plants was constructed, named R2. In 2020, according to the mapping results of R2, individual plants with heterozygosity in the target segment and homozygosity in the remaining background were selected from the population to construct another RHL population with 9 plants, named R3.

#### 4.1.3. Materials for Quantitative Real-Time PCR (qPCR)

In this study, SN14 was selected as the control material. From the R2 population, two plants D646-1 and D646-2 with SN14 background as homozygous target fragment, and two plants D646-3 and D646-4 with ZYD00006 background as homozygous target fragment were selected. The single plant was grown into a plant line, and 5 plant lines were obtained. According to the description of different soybean seed development stages by soybean base website (https://soybase.org/, accessed on 25 November 2018), the seeds of different plants in five rows were sampled at five growth and development stages of EM1, EM2, MM, LM and DS as quantitative analysis materials. Three biological replicates were performed for each row at each stage.

#### 4.1.4. Materials for Haplotype Analysis

A total of 350 soybean germplasm accessions collected from Heilongjiang Province, Jilin Province, Liaoning Province, and the Inner Mongolia Autonomous Region of China were used for haplotype analysis of the candidate gene (Appendix A).

### 4.2. Experimental Methods

#### 4.2.1. Methods for Planting and Field Management of Experimental Materials

From 2017 to 2020, the materials were grown at the Xiangyang Base of Northeast Agricultural University in Harbin (45.74° N, 126.73° E). The planting pattern was consistent across all years: 5 m row length, 60 cm row spacing, and 5 cm plant spacing, with unified management following local conventional field practices. During the growing-to-harvest period, the average temperatures over the four years were 17.66 °C, 17.51 °C, 17.21 °C, and 17.43 °C, respectively (Table 4); the average precipitation levels were 2.82 mm, 3.60 mm, 4.39 mm, and 4.20 mm, respectively (Table 5, Appendix A). The resource population used for haplotype analysis was grown at the Gongzhuling Base in Changchun from 2019 to 2021 (Appendix A).

#### 4.2.2. Determination of Soybean Seed Protein Content

The FOSS-1241TM Near-Infrared Grain Quality Analyzer was used to measure soybean seed protein content. During measurement, the moisture content of soybean seeds was maintained within the safe range. Each sample was measured in triplicate, and the average value of the three measurements was used as the final phenotypic data for seed protein content.

#### 4.2.3. DNA Extraction and Detection

Young leaf samples were collected from the top of the main stem in July and stored at −80 °C. Genomic DNA was extracted using the cetyltrimethylammonium bromide (CTAB) method, referring to the protocol described by Doyle J [42]. The purity and concentration of extracted DNA were detected using a Nanodrop spectrophotometer. DNA samples were considered qualified if the A_260_/A_280_ ratio ranged from 1.8 to 2.0 and the absorption curve showed a single peak.

#### 4.2.4. Acquisition and Identification of SSR Markers

Simple sequence repeat (SSR) markers were designed based on the Williams 82.a2.v1 reference genome sequence published on the Soybase website. Primers were synthesized by Shanghai Sangon Biotech Co., Ltd. The synthesized primers were in powder form: after centrifugation at 4000 rpm/min using a desktop centrifuge, the tube cap was carefully opened, and a corresponding volume of double-distilled water (ddH_2_O) was added to dissolve the primers according to the label. The tubes were sealed, vortexed thoroughly, and stored at −20 °C. Polyacrylamide gel electrophoresis (PAGE) was used to identify the polymorphism of SSR markers.

#### 4.2.5. Segregation Analysis of Traits (SEA)

The SEA software was used to predict the genetic pattern of quantitative trait genes in the biparental segregating population. The specific calculation formulas are as follows:

Probability of the Smirnov test (nW^2^):
P{W2n≤x}=12x∑k=0∞F(k+12)4k+1F(k+12)F(k+1)exp{−(4k+1)216x}×{I−14((4k+1)216x)−I−14((4k+1)216x)}

Probability of the Kolmogorov test (Dₙ):
$$\underset{n\to \infty }{\lim}P[\sqrt{n}D_{n}\leq \left.z\right|H_{0}]=K_{0}(z)+\frac{K_{1}(z)}{n^{1/2}}+\frac{K_{2}(z)}{n}+\frac{K_{3}(z)}{n^{3/2}}+O(\frac{1}{n^{2}})$$

#### 4.2.6. Single-Marker Analysis

Single-marker analysis was performed using the Single Marker Analysis module in Windows QTL Cartographer 2.5 software [43] to identify molecular markers significantly associated with seed protein content by analyzing phenotypic and genotypic data of the mapping population. A simple linear regression model was used for data analysis:
y=b0+b1x+e

#### 4.2.7. Functional Annotation of Candidate Genes

Genes within the candidate interval were annotated using the Williams 82.a2.v1 genome as the reference. The Phytozome database (https://phytozome.jgi.doe.gov/pz/portal.html, accessed on 20 February 2019) was used to analyze gene functions and predict candidate genes [44].

#### 4.2.8. Amino Acid Sequence Alignment of Candidate Genes

Based on the genotypic data from parental resequencing, variations (SNPs and Indels) in the 3000 bp upstream promoter region, 5′ untranslated region (UTR), CDS region, and 3′ UTR of candidate genes between the two parents were analyzed. The CDSs were translated into amino acid sequences to detect differences in amino acid sequences between the parents.

#### 4.2.9. Quantitative Real-Time PCR (qPCR) Analysis

Transcript sequences of candidate genes were obtained from the Phytozome website, and primers were designed using Primer 5 software. *GmActin4* (GenBank Accession No.: AF049106) was used as the reference gene. Total RNA was extracted using the Trizol method and reverse-transcribed into cDNA. qPCR was performed to analyze the relative expression levels of candidate genes.

#### 4.2.10. Promoter Element Analysis

A 3000 bp sequence upstream of the start codon (ATG) of candidate genes was defined as the promoter region, which was retrieved from the Phytozome database. The Plant CARE database (http://bioinformatics.psb.ugent.be/webtools/plantcare/html/, accessed on 28 June 2019) was used to identify and annotate cis-acting elements in the promoter region. TBtools software [45] was used for visualizing promoter elements.

#### 4.2.11. Haplotype Analysis of Candidate Genes in the Germplasm Population

Local BLAST analysis was performed to obtain SNP information of candidate genes (*Glyma.16G164900* and *Glyma.16G165100*) in the 350 soybean germplasm accessions. To investigate the genetic basis of seed protein content in soybean, haplotype analysis of candidate genes was conducted on 350 genotyped soybean accessions using CandiHap [46] (version 1.3.2). Polymorphic SNPs located in the upstream regulatory regions and exons were prioritized for haplotype construction. The phenotypic effects of distinct gene haplotypes were visualized using the R programming language (version 4.2.2) to elucidate their associations with seed protein content variation. Haplotypes with accessions accounting for more than 5% of the total population were defined as elite haplotypes. SPSS 26.0 software was used for one-way analysis of variance (ANOVA) to determine the effect of haplotypes on phenotypic variation. Independent samples t-tests were performed to compare protein content among different haplotypes, with statistical significance set at *p* < 0.05.

## 5. Conclusions

Based on a CSSL population, this study integrated genotypic data with phenotypic data of seed protein content to select elite lines for the construction of secondary segregating populations over consecutive years. Ultimately, a QTL associated with soybean seed protein content, designated as *qSPJ-1*, was mapped to a region on soybean chromosome 16 spanning 32,403,844 bp to 32,480,066 bp. Within this interval, through sequence alignment, expression level analysis, and haplotype analysis, the WD-40 protein family gene *Glyma.16G165100* was preliminarily predicted as a potential candidate gene influencing soybean seed protein content. The identified superior haplotypes Hap_3 and Hap_4 exhibited higher average protein content. Breeders can develop molecular markers targeting these two superior haplotypes for use in marker-assisted selection breeding to genetically improve related traits. This study utilized wild soybean resources to construct the mapping population, enabling the detection of rare elite allelic variants derived from wild resources. Notably, landraces with wild soybean ancestry were included in Hap_3 and Hap_4, which provides valuable insights for the effective utilization of wild soybean resources in the genetic improvement of soybean protein content and offers potential to enrich the genetic basis of cultivated soybeans in China and globally. Future work will focus on verifying the function of *Glyma.16G165100* through genetic transformation and investigating its applicability across different environments.

## Figures and Tables

**Figure 1 plants-14-03525-f001:**
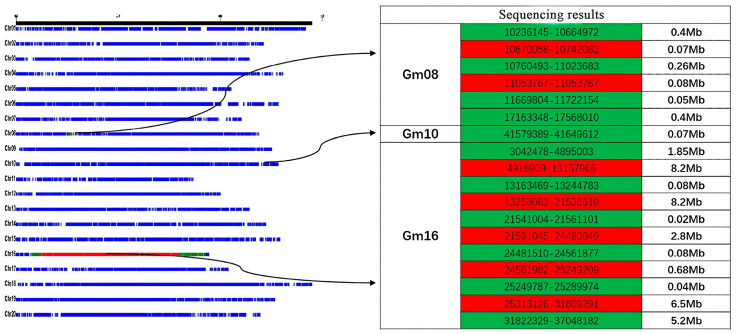
CSSL-646 resequencing BinMap results. Note: Blue represents the background segment; green represents the homozygous import segment; red represents the heterozygous segment.

**Figure 2 plants-14-03525-f002:**
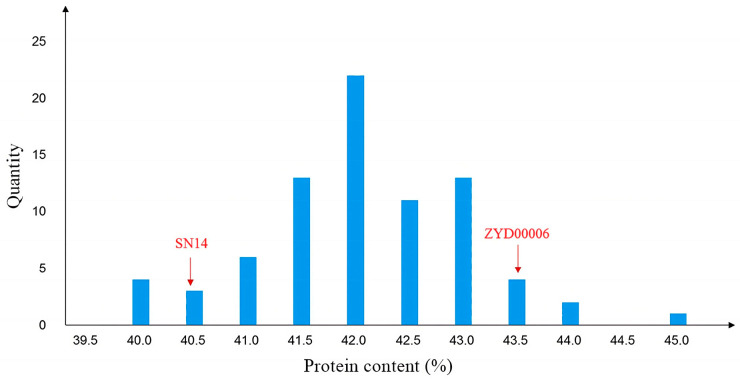
The protein content phenotypic distribution of F2.

**Figure 3 plants-14-03525-f003:**
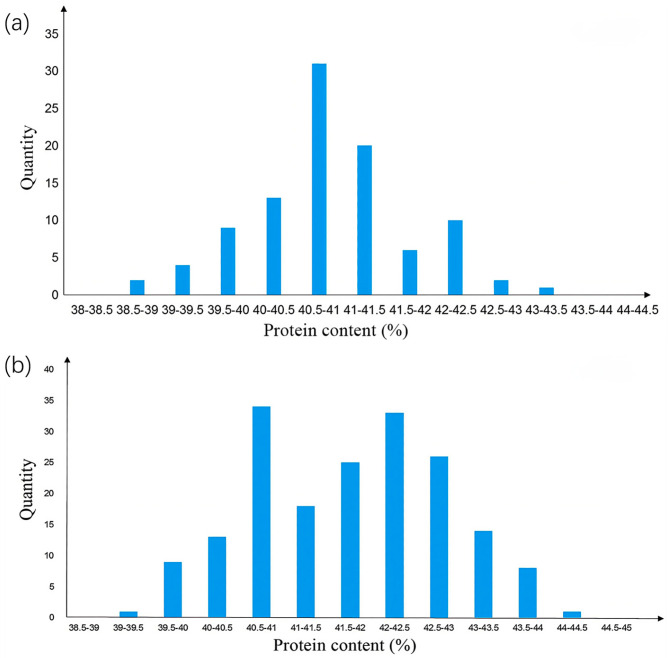
Protein content distribution histogram of secondary segregating population. Note: (**a**) The Phenotypic Distribution of Protein Content in the R1 Population. (**b**) The Phenotypic Distribution of Protein Content in the R2 Population.

**Figure 4 plants-14-03525-f004:**
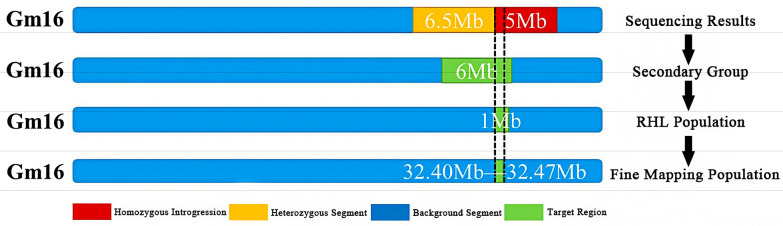
Research process of Gm16 fine positioning.

**Figure 5 plants-14-03525-f005:**
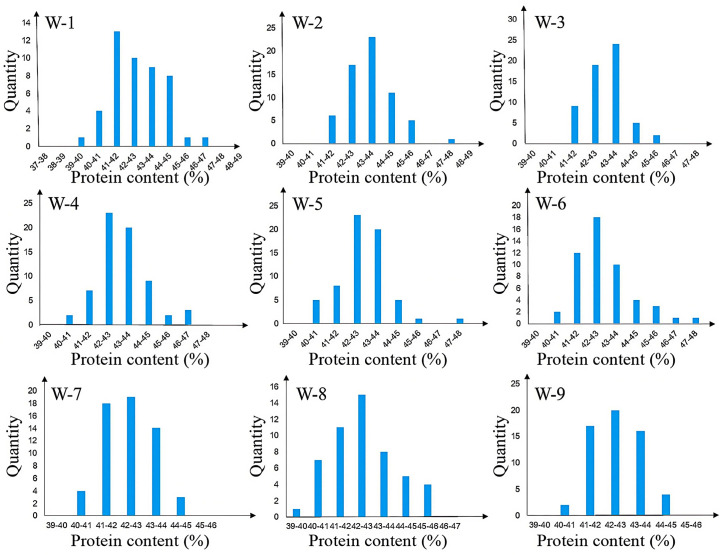
Fine mapping of protein phenotypic distribution of population.

**Figure 6 plants-14-03525-f006:**
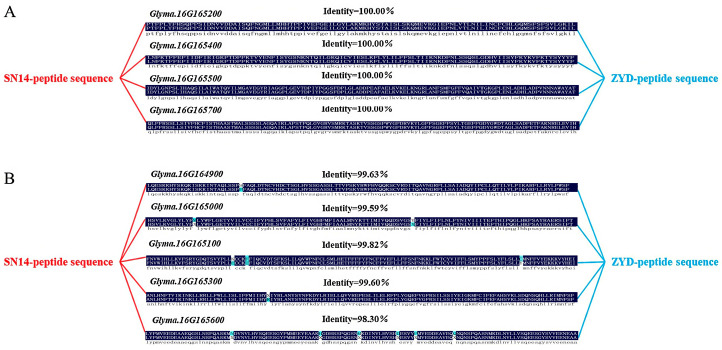
Amino acid changes in candidate genes in parents. Note: (**A**): Amino acid sequence does not change between parents; (**B**): Amino acid sequence is different between parents.

**Figure 7 plants-14-03525-f007:**
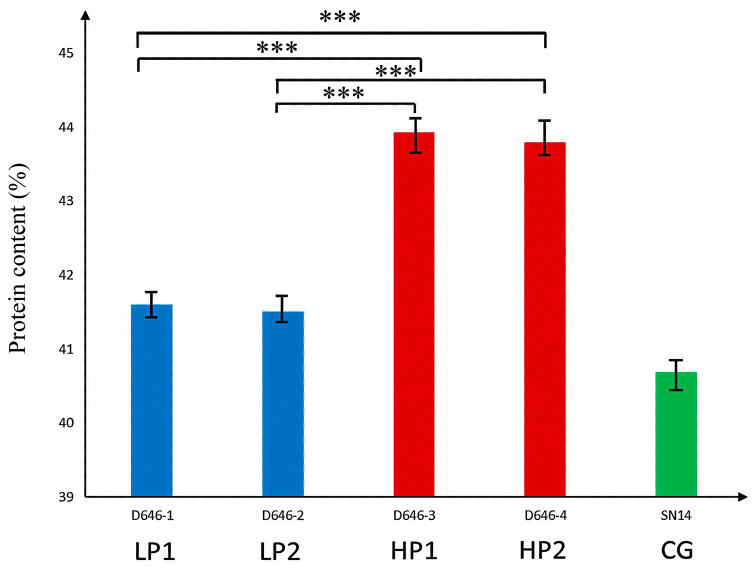
Real-time quantitative analysis of the significance of protein phenotypic differences in extreme materials. Note: *** *p* ≤ 0.001.

**Figure 8 plants-14-03525-f008:**
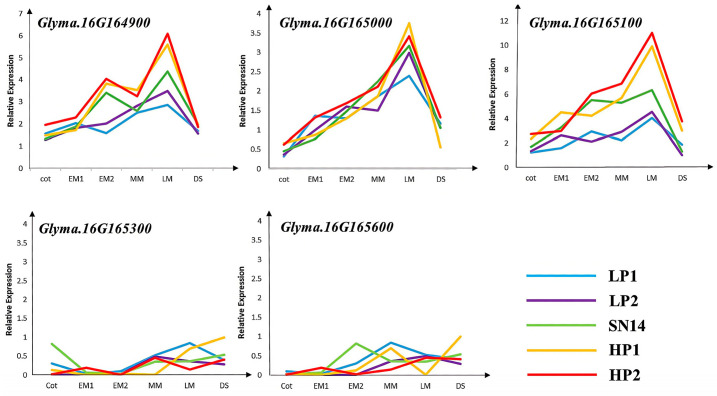
Specific expression analysis of candidate genes at different stages of seed development. Note: Green represents parent SN14; blue and purple represent high protein materials; yellow and red represent low protein materials.

**Figure 9 plants-14-03525-f009:**
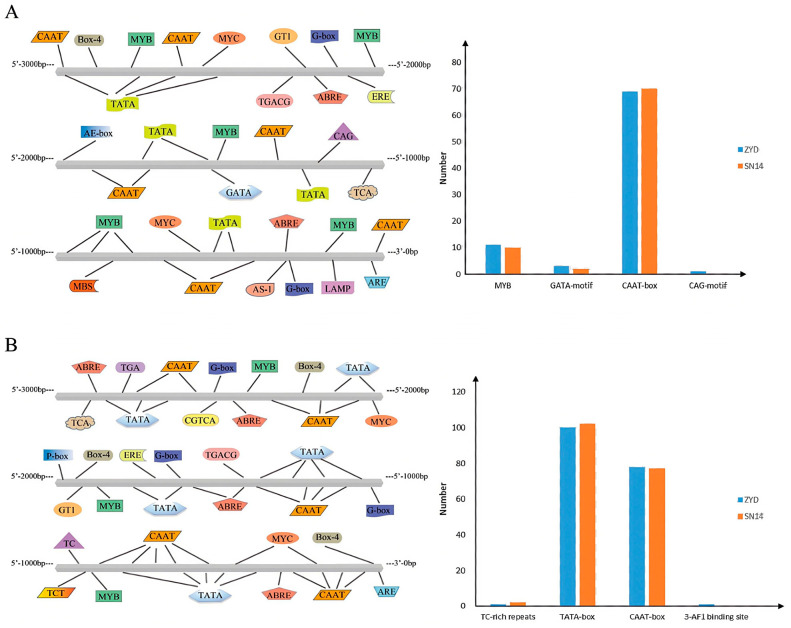
Promoter element analysis of candidate genes. Note: (**A**): *Glyma.16G164900* promoter element analysis and parent-to-parent comparison; (**B**): Promoter element analysis and parent-to-parent comparison of *Glyma.16G165100*.

**Figure 10 plants-14-03525-f010:**
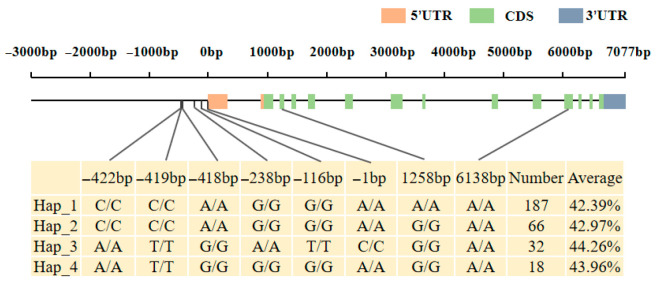
Distribution Map of Haplotype Variation Sites of Candidate Gene *Glyma.16G165100*.

**Figure 11 plants-14-03525-f011:**
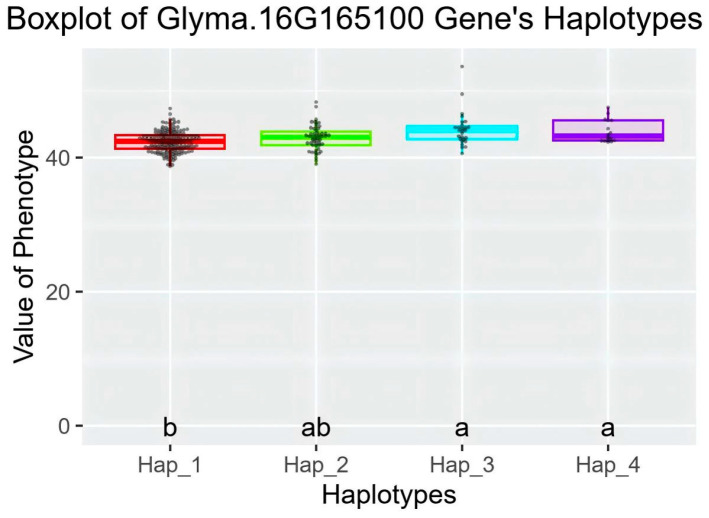
Boxplot of *Glyma.16G165100* Gene’s Haplotypes.

**Table 1 plants-14-03525-t001:** Segregation analysis of soybean protein about F2.

Population	Model	AIC	M-G Var	Heritability (M-G)	P (nW)	P (Dn)
F_2_	1MG-A	225.7	0.091	0.094	0.884	1

Note: M-G Var: Major Gene Variance; Heritability (M-G): Major Gene Heritability; P (nW): Probability of the nW^2^ statistic in the Smirnov test; P (Dn): Probability of the Dn statistic in the Kolmogorov test.

**Table 2 plants-14-03525-t002:** Segregation analysis of soybean protein about RHL.

Population	Model	AIC	M-G Var	Heritability (M-G)	*nW^2^*P (nW)	P (Dn)
F_2_(R1)	1MG-A	275.6	0.024	0.026	0.265	1
F_2_(R2)	1MG-AD	542.774	0.866	0.736	0.914	1

Note: M-G Var: Major Gene Variance; Heritability (M-G): Major Gene Heritability; P (nW): Probability of the nW^2^ statistic in the Smirnov test; P (Dn): Probability of the Dn statistic in the Kolmogorov test.

**Table 3 plants-14-03525-t003:** Candidate gene annotation information.

Gene Name	Homologous Gene	GO Annotation	KEGG Annotation	nr Annotation
*Glyma.16G164900*	*AT1G76730.1*	none	none	NAGB/RpiA/CoA Transferase Superfamily Protein
*Glyma.16G165000*	*AT1G43190.2*	GO:0003676	K14948	Polypyrimidine Tract-Binding Protein
*Glyma.16G165100*	*AT5G52820.1*	GO:0005515	K14855	WD-40 Repeat Family Protein
*Glyma.16G165200*	*AT2G34430.1*	GO:0016020	K08912	Light-Harvesting Chlorophyll Protein Complex II Subunit B1
*Glyma.16G165300*	*AT1G34040.1*	GO:0016846	none	Pyridoxal 5′-Phosphate Transferase Superfamily Protein
*Glyma.16G165400*	*AT1G12300.1*	none	K17710	Tetratricopeptide Repeat Superfamily Protein
*Glyma.16G165500*	*AT2G34430.1*	GO:0016020	K08912	Light-Harvesting Chlorophyll Protein Complex II Subunit B1
*Glyma.16G165600*	*AT1G76740.1*	none	none	none
*Glyma.16G165700*	*AT5G10090.1*	none	none	Tetratricopeptide Repeat Superfamily Protein

Note: GO annotation: Gene Ontology annotation; KEGG annotation: Kyoto Encyclopedia of Genes and Genomes annotation; nr annotation: Non-Redundant Protein Database annotation.

**Table 4 plants-14-03525-t004:** Weather Conditions in Changchun from May to October.

	Maximum Temperature	Minimum Temperature	Average Temperature	Active Accumulated Temperature	Daily Average Active Accumulated Temperature
2017	35 °C	−6 °C	17.66 °C	3986.5 °C	26.2 °C
2018	35 °C	−3 °C	17.51 °C	4129.5 °C	26.1 °C
2019	33 °C	−4 °C	17.21 °C	3876.0 °C	26.5 °C
2020	33 °C	−4 °C	17.43 °C	4329.0 °C	26.7 °C

**Table 5 plants-14-03525-t005:** The precipitation situation from May to October.

	Maximum Precipitation	Minimum Precipitation	Average Precipitation
2017	29.87 mm	0 mm	2.82 mm
2018	37.50 mm	0 mm	3.60 mm
2019	46.89 mm	0 mm	4.39 mm
2020	55.92 mm	0 mm	4.20 mm

## Data Availability

All data are contained within the article.

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
