# Peer review of "Fine Mapping of *qSPJ_1* and Candidate Gene Identification for Soybean Seed Protein Content"

_plants, 2025, doi:10.3390/plants14223525_

Round 1

Reviewer 1 Report

Comments and Suggestions for Authors

The current version of abstract is narrative and should be reformatted accordingly.

Although the work identifies a promising candidate gene, the discussion does not sufficiently highlight the novelty compared with previous QTL studies (e.g., Zhang et al. 2017; Qi et al. 2024; Yang et al. 2025). The authors should emphasize what is unique in their approach (CSSL-based fine mapping, haplotype validation).

The Discussion section is mainly descriptive. It should provide a critical comparison with earlier fine-mapping efforts on Gm16 and related WD-40 proteins.

Limitations should be acknowledged: environmental validation, number of populations, and possible pleiotropic effects. The breeding implications (marker-assisted selection, CRISPR targeting of Glyma.16G165100) should be expanded.

Many recent references (2022–2025) are cited but not deeply integrated into the discussion. This weakens the context of the findings.

Figures need to follow MDPI format: Titles above the figure, notes below. Higher resolution for Figures 3–5 (currently low clarity in phenotypic distributions). Sub-panels (A and B) should be clearly labeled within the figures. Tables require consistent styling (e.g., Table 1–3 should include clear footnotes).

Abbreviations should be defined at first use and compiled in the “Abbreviations” section (CSSL, QTL, SNP, RIL, etc.).

Methods section: software tools (SEA, SPSS, TBtools, Phytozome) should be cited with version numbers and official references/URLs.

Promoter analysis: include statistical or visual evidence to strengthen the claim that promoter variations are regulatory.

Clarify environmental growth conditions for mapping populations (year, location, field/in greenhouse).

Reviewer 2 Report

Comments and Suggestions for Authors

Dear Authors,

Your work involves a lot of labor and effort. It is an original piece of work, but I would like to make some suggestions. I believe that addressing these shortcomings would make the work more meaningful.

1-I believe it would be better to change the title to “Fine Mapping of qSPJ_1 and Candidate Gene Identification for Soybean Seed Protein Content.”

For the Introduction;

1- The introduction contains a lot of information, but the paragraphs are long and the reader's attention may wander. It would be more readable if divided into subheadings (e.g., “Importance of soy protein,” “Environmental & genetic factors,” “Progress in molecular breeding”).

2- Most of the studies are cited from Chinese researchers. Sources from a broader international literature (major soybean research centers such as the US, Brazil, Europe) could be added.

3- Several recent references from GWAS/QTL studies published in the last 3 years should be added.

4- The research gap is not clear: The summary of previous studies is good, but the section “What gap does this article fill?” is not sufficiently clear.

5- It only states that “we used CSSL and obtained more accurate results.” This should be stated more clearly.

For Material and Methods;

1- Populations and marker selections are described, but the number of replicates, statistical power analysis, or limitations of the experimental design are unclear. In particular, environmental variation (year × location effect) is not mentioned. Were all experiments conducted at a single location?

2- Control Groups:

The samples used in qPCR analysis are limited (only 4 individuals + control). It is unclear whether there were broader biological replicates.

3- Lack of References for Methods:

A reference to Doyle (1990) is provided for CTAB DNA extraction, but most other methods lack source citations. E.g., literature support should be added for SEA software, QTL Cartographer, and SPSS analyses.

4- Haplotype Analysis:

It is stated that 336 germplasm were used, but the geographical/ecological distribution of these varieties is not explained. This limits the generalizability of haplotype results to larger populations.

5- Failure to Specify Environmental Conditions:

The location and year(s) of the trials, as well as climate/environmental conditions, are not provided. Since protein contents are affected by environmental factors, this omission represents a major methodological weakness.

For Results;

1- Data Repetition and Reliability:

qPCR experiments were conducted with only 4 individuals. This reduces the statistical reliability of the results. Wider biological repetitions (different locations and years) should be presented.

2- Overinterpretation of Results:

For example, highlighting Glyma.16G165100 as a “key candidate gene” is very bold. Since functional validation (e.g., knockout/overexpression experiments) has not yet been performed, this statement is premature. More cautious language (“strong candidate” or “likely candidate”) should be used.

3- Lack of Haplotype Analysis:

336 accessions were used, but the geographical distribution and diversity of these accessions were not explained. The comparison of haplotype frequencies in different ecological regions was not performed.

For Discussion;

1- Overly Ambitious Presentation of Results:

The authors present Glyma.16G165100 as a “key candidate gene” in almost definitive terms. However, this is based solely on mapping + haplotype + qPCR data; there is no functional validation (e.g., CRISPR knockout, overexpression, transgenic tests). Therefore, the results should be presented in more cautious language.

2- Environmental Factors Excluded:

The discussion does not address the possible contributions of environmental effects (climate, soil, location differences) to candidate gene expression and protein content. Findings are reduced solely to genetic factors.

3- Lack of Limitations:

The limitations of the study are not clearly stated in the discussion section. For example: Single location and limited trial year, Few qPCR replicates, Lack of functional validation, Limited explanation of germplasm diversity. Because these are not mentioned, the section remains somewhat “one-sidedly emphasizing strong results.”

4- Lack of International Context:

Literature comparisons are mostly based on Chinese studies. The discussion would have gained universality if examples from major soybean production regions such as Brazil and the US had been included.

5- Protein–Yield Correlation:

The “negative correlation between protein content and yield” mentioned in the introduction was not revisited in the discussion section. This connection would have strengthened the practical significance of the study.

For the Conclusion;

1- Overly Ambitious Results:

Glyma.16G165100 was directly presented as “the key candidate gene.” Since functional validation (e.g., CRISPR/Cas9 knockout, transgenic tests) was not performed, this claim should have been made more cautiously. For example: “Glyma.16G165100 is a strong candidate gene and requires further functional validation.”

2- Failure to Mention Limitations:

The fact that the experiment was conducted at a single location and for a short period, the limited number of qPCR samples, and the failure to consider environmental factors were not discussed at all. At least a brief sentence mentioning these limitations in the conclusion section would have been expected. 

3- Insufficient Explanation of Practical Contribution:

The conclusion emphasizes “theoretical guidance,” but it is unclear how it would contribute to breeding programs in practice. For example: Its applicability for marker-assisted selection (MAS), how the developed haplotypes would be evaluated by breeders, and which stages of the variety development process it would contribute to could have been added.

4- Lack of Broader Context:

The study's significance for global soybean breeding or protein quality research is not provided. Adding a more international context would make the results appeal to a wider audience.

Reviewer 3 Report

Comments and Suggestions for Authors

This manuscript “Fine Mapping and Candidate Gene Mining of qSPJ_1 Control-2 ling Seed Protein Content in Soybean” based on previous whole-genome introgression line population, constructed a secondary segregating populations to fine-map QTLs for soybean seed protein content, to identify a candidate gene Glyma.16G165100 as a key gene influencing soybean seed protein content. The author selected four individuals with heterozygous genotypes to further functional map 0.076 Mb which identified 9 candidate genes. After further using qRT-PCR, promoter element analysis, haplotype analysis, the author thus identified Glyma.16G165100 was the key candidate gene controlling soybean seed protein content. All the data the author conclude the Glyma.16G165100 was the candidate gene, which missing the critical genetic data to support the gene real function. From the gene annotation which Glyma.16G165100 was WD-40 repeat family proteins, the author should have more results or at lease discussion how this gene to regulate or involve in the soybean seed protein content. From the results section, the figure legends are too simple to fully understand each figure content. The methods section also need more detailed information. The discussion section is needing more detailed discussion to the possible mechanisms of how Glyma.16G165100 regulate the soybean seed protein content. The references sections need to be revised carefully each by each which most of them are not matched correctly.  

Round 2

Reviewer 1 Report

Comments and Suggestions for Authors

The authors have addressed the suggested observations, considerably enriching the document to provide potential readers with reproducible results and greater argumentative quality.

Reviewer 2 Report

Comments and Suggestions for Authors

Dear Authors,

The changes you made based on our suggestions have made your article even more meaningful.